# Activation of neuronal genes via LINE-1 elements upon global DNA demethylation in human neural progenitors

Marie E Jönsson[1], Per Ludvik Brattås[1], Charlotte Gustafsson[2], Rebecca Petri[1], David Yudovich[3], Karolina Pircs[1], Shana Verschuere[1], Sofia Madsen[1], Jenny Hansson [4], Jonas Larsson[3], Robert Månsson[2], Alexander Meissner[5] & Johan Jakobsson [1]

DNA methylation contributes to the maintenance of genomic integrity in somatic cells, in part through the silencing of transposable elements. In this study, we use CRISPR-Cas9 technology to delete *DNMT1*, the DNA methyltransferase key for DNA methylation maintenance, in human neural progenitor cells (hNPCs). We observe that inactivation of *DNMT1* in hNPCs results in viable, proliferating cells despite a global loss of DNA CpG-methylation. DNA demethylation leads to specific transcriptional activation and chromatin remodeling of evolutionarily young, hominoid-specific LINE-1 elements (L1s), while older L1s and other classes of transposable elements remain silent. The activated L1s act as alternative promoters for many protein-coding genes involved in neuronal functions, revealing a hominoid-specific L1-based transcriptional network controlled by DNA methylation that influences neuronal protein-coding genes. Our results provide mechanistic insight into the role of DNA methylation in silencing transposable elements in somatic human cells, as well as further implicating L1s in human brain development and disease.

[1] Wallenberg Neuroscience Center and Lund Stem Cell Center, Laboratory of Molecular Neurogenetics, Department of Experimental Medical Science, BMC A11, Lund University, 221 84 Lund, Sweden. [2] Center for Hematology and Regenerative Medicine Huddinge, Karolinska Institute, 141 52 Stockholm, Sweden. [3] Division of Molecular Medicine and Gene Therapy, Department of Laboratory Medicine and Lund Stem Cell Center, BMC A12, Lund University, 221 84 Lund, Sweden. [4] Laboratory of Proteomic Hematology, Department of Laboratory Medicine and Lund Stem Cell Center, BMC B12, Lund University, 221 84 Lund, Sweden. [5] Department of Genome Regulation, Max Planck Institute for Molecular Genetics, 14195 Berlin, Germany. Correspondence and requests for materials should be addressed to J.J. (email: johan.jakobsson@med.lu.se)

More than 50% of the human genome is derived from mobile genetic elements, the majority of which are retrotransposons that have colonized genomes throughout evolution by copy-and-paste mechanisms[1]. Transposable elements (TEs) pose a threat to genomic integrity and are therefore mostly transcriptionally silenced in the earliest stages of embryonic development[2]. However, transposition may also be beneficial for the host since new integration events may, for example, create novel and beneficial expression patterns of nearby protein-coding genes[3]. LINE-1 elements (L1s), which are non-LTR transposons, can transpose during brain development and in neural progenitor cells (NPCs), a process which may contribute to intra-individual variation in brain function[4–10]. In addition, we have previously found that TEs influence transcriptional networks in NPCs by acting as hubs for epigenetic marks[11,12], and several studies have linked aberrant transcriptional activation of TEs to brain disorders[13–16].

In normal somatic tissues, the vast majority of TEs are transcriptionally repressed, which correlates with the presence of DNA CpG-methylation[17]. The establishment of DNA methylation on TEs, which occurs early in development, is thought to be essential for attracting silencing complexes that repress the transcription of TEs in somatic cells[18]. However, causative data and mechanistic insights into these processes are limited: deletion of DNA methyltransferase 1 (DNMT1), the enzyme that maintains DNA methylation during cell division, is lethal in all dividing somatic mouse cells as well as human cancer cells[19–21].

In this study, we disrupt DNMT1 in human NPCs (hNPCs) using CRISPR-Cas9 based gene editing. Despite a global loss of CpG-methylation, hNPCs lacking DNMT1 activity survive and remain proliferative, thereby providing an excellent model system for investigating the role of DNA methylation in the control of TEs in somatic cells. We find that loss of DNA methylation results in transcriptional activation of evolutionarily young, full-length L1s, coupled with the acquisition of the active histone mark H3K27ac. Older L1s, as well as other classes of TEs, are not affected. The activated L1s serve as alternative promoters for many nearby protein-coding genes previously identified as being involved in neuronal functions and psychiatric disorders. These results provide a fundamental mechanistic insight into the role of DNA methylation in transcriptional control of TEs in human somatic cells and describe a mechanism for how L1 activation influences human brain development and related disorders through its impact on gene regulatory networks.

## Results

**Targeted disruption of DNMT1 in hNPCs.** To provide a functional insight into the role of DNA methylation in controlling the expression of TEs during human brain development, we used CRISPR-Cas9 technology to delete DNMT1 in hNPCs. We used a previously described lentiviral Cas9-GFP-gRNA construct that expresses a Cas9-GFP transcript, and a gRNA that targets the catalytic domain of DNMT1 in exon 32[22]. When used in human embryonic stem cells, this vector results in a complete loss of DNMT1 activity, a global loss of CpG methylation, and rapid cell death[22]. We transduced the fetal-derived hNPC line Sai2, which has the characteristics of neuroepithelial-like stem cells[23], with LV.CRISPR-DNMT1 followed by FACS-GFP isolation 10 days later (Fig. 1a). As the control, we used a similar lentiviral vector, but with a gRNA directed toward LacZ.

To assess for functional inactivation of DNMT1, we performed immunocytochemical staining for 5-methylcytosine (5mC) on the transduced cultured cells. On day 10, we found that more than 90% of the sorted, LV.CRISPR-DNMT1 transduced hNPCs lacked any detectable 5mC, while control cells displayed a distinct, constant 5mC signal, demonstrating that the enzymatic activity of DNMT1 was efficiently disrupted using this approach (Fig. 1b, Supplementary Fig. 1a and Supplementary Fig. 2a). It is worth noting that DNMT1 mRNA levels in these cells were only reduced by about 50% following LV.CRISPR-DNMT1 transduction (Supplementary Fig. 1b), which is consistent with the generation of small insertions/deletions (indels) that disrupt protein levels and function (as expected), but not the presence of the mRNA. The levels of DNMT1 protein were, on the other hand, greatly reduced (Supplementary Fig. 1c).

**Lack of DNMT1 activity is compatible with viable hNPCs.** Despite the global loss of DNA methylation, the transduced, cultured hNPCs remained viable. This was surprising, since previous attempts to generate somatic cells without DNMT1, in both mouse and human, resulted in non-viable cells[19,20]. We therefore performed a series of control experiments to substantiate our findings. To verify that there were no contaminating cells of embryonic origin in the hNPC line, we confirmed that the cells all had the same DNA methylation level, as assessed by 5mC staining, morphology, and the expression of NPC marker genes such as NES and SOX2 (Fig. 1b, Supplementary Fig. 1d–e). We also verified that the morphology and expression of NES and SOX2 in DNMT1-KO hNPCs were similar to control hNPCs, despite displaying no 5mC staining (Supplementary Fig. 1d–e).

To confirm that the DNMT1-KO cells were not the result of a selective clonal expansion of a rare, transformed subpopulation of cells, we analyzed the amount and nature of indels in the cells 10 days after transduction through PCR-amplification of genomic DNA flanking the gRNA target site, followed by next-generation sequencing. Since CRISPR-Cas9-mediated gene disruption results in imperfect and random non-homologous end joining repair, the resulting indels should be highly variable. There should be deletions and insertions of different sizes unless clonal selection occurred. When analyzing two independent rounds of LV.CRISPR-DNMT1 transduced hNPCs, we found that $93.5 \pm 3.8\%$ of the targeted sequence in DNMT1-KO hNPCs were modified and had highly variable indels 10 days post-transduction (unique frameshift indels $n = 222 \pm 85$, unique in-frame indels $n = 136 \pm 49$): this makes it highly unlikely that there was any clonal selection.

To verify that DNMT1-KO hNPCs remained proliferative, we performed BrdU pulsing experiments at different time-points (days 10–15) after LV.CRISPR-DNMT1 transduction. These pulsing experiments were followed by dual immunocytochemistry analysis for BrdU and 5mC. We found that hNPCs incorporated BrdU despite the lack of 5mC staining at all time-points investigated, demonstrating that DNMT1-KO hNPCs remained proliferative for at least 15 days without DNA methylation (Fig. 1c, Supplementary Fig. 1f). Quantification of the proportion of BrdU-incorporating cells confirmed that a large proportion of the DNMT1-KO cells remained proliferative, although they divided at a slower rate than control cells (Supplementary Fig. 1f).

By performing RNA-seq analysis and in-depth proteomics, we also found that a small number of protein-coding genes that are known to be controlled by CpG methylation were activated upon demethylation (Supplementary Fig. 1g–i, Supplementary Data 1). These genes included germ-line restricted genes such as MAEL and DAZL, imprinted genes such as NNAT, H19, and TFPI2, as well as the lncRNA XIST (Supplementary Fig. 1j–k).

We also verified these results by performing LV.CRISPR-DNMT1 transduction on an additional hNPC cell line which was derived from human induced pluripotent cells (iPSCs). These iPSC-derived hNPCs also survived the loss of DNA methylation, continued to divide and showed an upregulation of similar genes

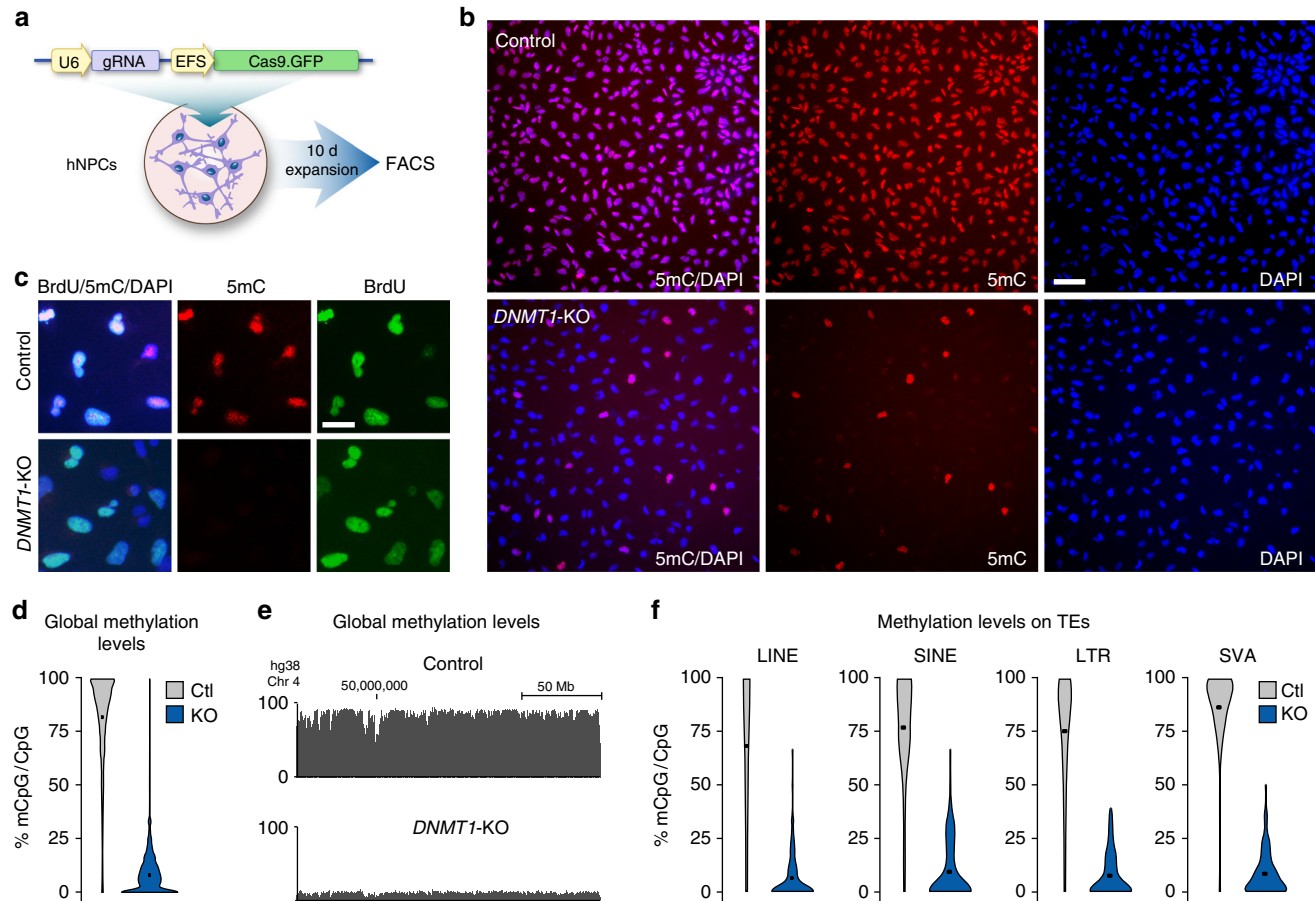

**Fig. 1** Disruption of *DNMT1* in hNPCs leads to global loss of DNA methylation. **a** Schematic workflow of generation of *DNMT1*-KO hNPCs using LV.CRISPR-DNMT1. **b** Fluorescent immunostaining of 5mC 10 days post transduction with LV.CRISPR-DNMT1 and the control vector, scalebar 70 µm. **c** Control and *DNMT1*-KO hNPCs were allowed to incorporate BrdU between days 10–15 post transduction. Fluorescent immunostaining of 5mC and BrdU revealed that *DNMT1*-KO cells continued to divide and incorporate BrdU even after loss of DNA methylation, scalebar 25 µm. **d** The distribution of genome-wide mCpG/CpG ratios of 1 kb tiles is shown by Violin plots. Each data point is the mean mCpG/CpG ratio within each 1 kb tile. The black dot shows the mean of all 1 kb bin means. **e** UCSC screenshot of a part of chromosome 4 shows global loss of CpG methylation upon loss of DNMT1. **f** Violin plots show distribution of mCpG/CpG within all LINE, SINE, LTR, and SVA elements in hg38. The black dot shows the mean of all elements

(Supplementary Fig. 1l). Taken together, these data demonstrate that hNPCs with no *DNMT1* activity survived and remained proliferative for at least 15 days in culture.

**Global DNA demethylation upon inactivation of *DNMT1* in hNPCs.** To investigate the consequences of *DNMT1* inactivation on DNA methylation levels more precisely, we performed whole-genome bisulfite sequencing (WGBS) on *DNMT1*-KO and control hNPCs. We found that control hNPCs displayed a high level of DNA methylation (79.0% of CpGs methylated, Fig. 1d–e), including high levels of methylation on various classes of transposable elements, such as LINE-1 (81.8%), LTR (79.5%), SINE (87.4%), and SVA (86.5%) (Fig. 1f). *DNMT1* inactivation resulted in a massive global loss of CpG methylation, with only 7.9% of CpGs methylated in LV.CRISPR-DNMT1 cells (Fig. 1d–e). There was also a global loss of DNA methylation on different classes of TEs: LINE (7.6%), LTR (7.1%), SINE (8.7%), and SVA (8.5%) (Fig. 1f). On the other hand, we found that non-CG DNA methylation was unaffected in *DNMT1*-KO cells (0.4% in both control and KO cells), which is consistent with the established role of the de novo DNA methyltransferases DNMT3A/B. These control non-CG DNA methylation, a process that is independent of DNMT1[22].

**Upregulation of L1 mRNA and protein upon DNA demethylation.** To characterize the expression of TEs after disruption of *DNMT1* in hNPCs, we performed 2 × 150 bp paired-end, strand-specific RNA sequencing and mapped the reads to the human reference genome (hg38). We discarded all reads that mapped equally well to multiple loci, hence keeping only reads that mapped uniquely to a single locus. Out of 4,168,182 TEs belonging to SINE, LINE, LTR, or SVA elements, we detected 45,442 elements with at least five reads. In all, 2,152 individual TEs were significantly upregulated (*p*-adj < 0.05), while only 72 were downregulated upon loss of DNMT1 (only TEs annotated in RepeatMasker (hg38) and non-overlapping NCBI gene exons were included in the analysis) (Fig. 2a). Strikingly, the majority of significantly upregulated TEs belonged to different L1 families (Fig. 2b, Supplementary Fig. 1m–n) as well as the LTR family LTR12C (Fig. 2b, Supplementary Fig. 1o). Since L1s have colonized the human genome via a copy-and-paste mechanism in different waves during evolution, it is possible to approximate the evolutionary age of each L1 integrant and assign them to chronologically ordered subfamilies. By exploiting this feature, we found that upregulated L1s belonged to lineages younger than 12.5 million years (L1HS, L1PA2, and L1PA3 subfamilies), therefore corresponding to hominoid-specific elements, including many that are human-specific (Fig. 2c).

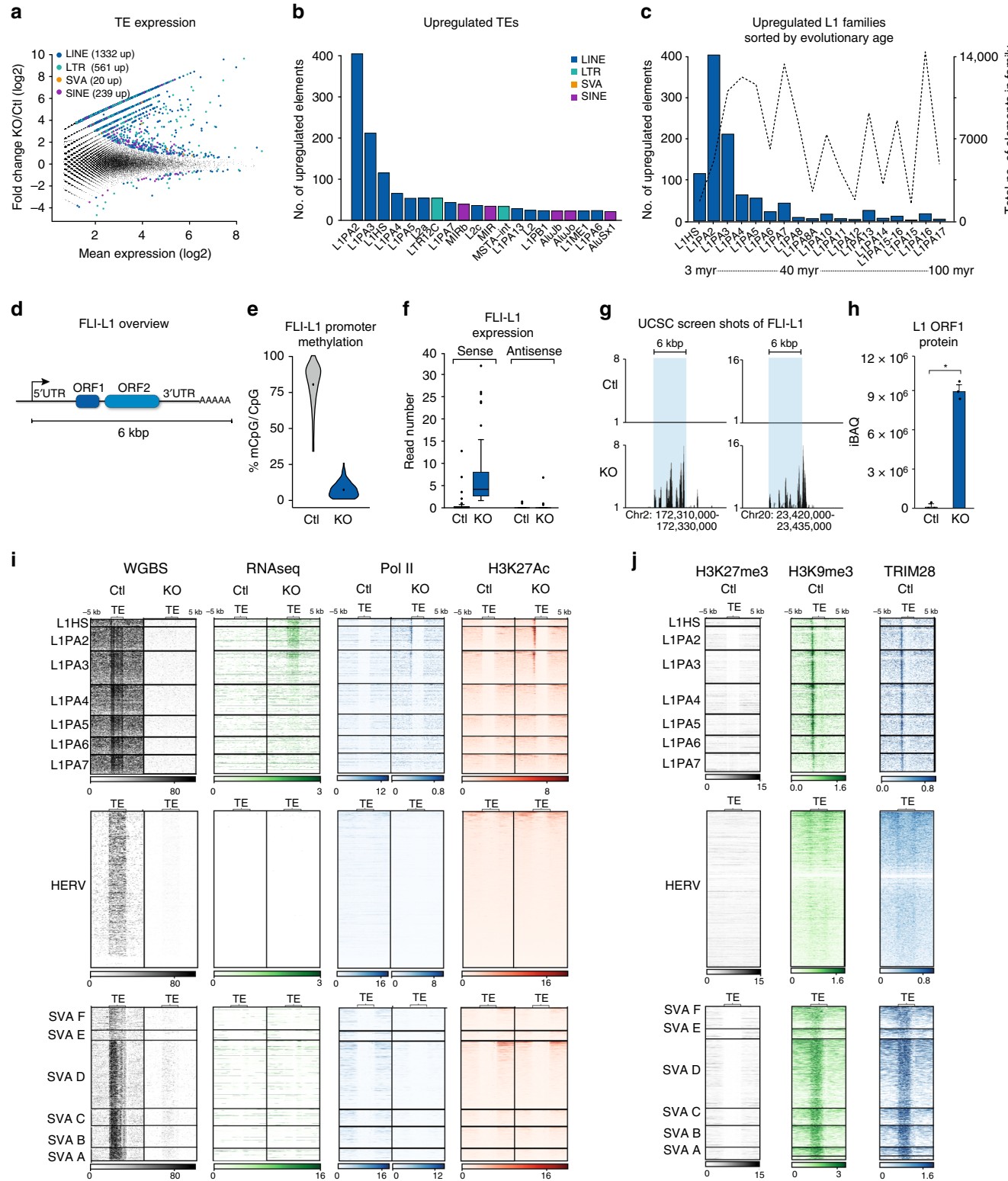

The human genome holds around half a million individual copies of L1s, including ancient fragments and more recent full-length copies. Most L1s are inactivated due to truncations and accumulation of inactivating mutations, and only a small number remain capable of retrotransposition[24,25]. In theory, L1s that are full length (>6kbp) with intact open reading frames (ORF) 1 and 2, 5′UTR internal promoter, and 3′UTR region, have the capacity to retrotranspose. We refer to these as full-length intact L1s (FLI-L1) (Fig. 2d)[26]. We found that these had high levels of CpG methylation (79.9%) in hNPCs, and that this methylation was lost upon *DNMT1*-KO (6.5%, Fig. 2e).

In accordance with the observed reduction in CpG methylation, we detected robust transcriptional activation of virtually all FLI-L1s following *DNMT1*-KO (Fig. 2f–g). The evolutionarily youngest L1s have a reduced mapability due to a very high sequence similarity. However, most copies have several unique SNPs relative to each other and to the consensus sequence, allowing any given young L1 to be discriminated from similar

**Fig. 2** Primate-specific L1s are activated upon global demethylation in hNPCs and Pol II and H3K27Ac were deposited on L1 promotors. **a** The fold change of expression of all individual TEs in the human genome upon *DNMT1*-KO. Mean expression is taken from all samples of both conditions ($n = 3$ in both groups). Significance threshold at BH-corrected $p < 0.05$. **b** The number of upregulated elements per TE family in the *DNMT1*-KO ($n = 3$) vs control ($n = 3$). The top 20 families with the highest number of upregulated elements are shown. **c** The L1 families are arranged based on evolutionary age: the average number of upregulated elements in each family upon *DNMT1*-KO and the total number of elements in each family are plotted on the left and right y-axes, respectively ($n = 3$ in both groups). **d** A schematic overview of an FLI-L1. **e** Distribution of CpG methylation (mCpG/CpG) at FLI-L1s ($n = 145$). Each data point is the mean mCpG/CpG within an FLI-L1. The black dot shows the mean of all elements. **f** The expression of FLI-L1s in the sense and antisense directions shown as boxplots. The expression in the sense direction was significantly upregulated, $p < 2.2e{-}16$, two-sided Wilcoxon rank sum test. The boxplot elements represent: center line, median; box limits, upper and lower quartiles; whiskers, ×1.5 interquartile range; dots, outliers. **g** Screenshots from UCSC showing two examples of FLI-L1s (indicated by blue boxes) that are upregulated in the *DNMT1*-KO hNPCs. **h** The proteomic analysis detected L1 ORF1 protein in the *DNMT1*-KO hNPCs. Error bars represent SEM from $n = 3$ per group, $p = 0.0019$, student's *t*-test. Source data is provided as a Source Data file. **i** Heatmaps of WGBS, RNA-seq, ChIP-seq of Pol II and H3K27ac in control and *DNMT1*-KO hNPCs in the eight youngest L1 subfamilies, all FL-HERVs, and all SVAs. WGBS data are shown as % mCpG at all CpG sites. RNA-seq is shown as sum of all three replicates at each base. ChIP-seq of Pol II and H3K27ac is shown as mean signal of both replicates at each base. **j** Heatmaps of ChIP-seq signals of TRIM28, H3K9me3, and H3K27me3 in hNPCs in the eight youngest L1s, FL-HERVs, and SVAs. ChIP-seq of TRIM28 and H3K9me3 is shown as log2ratio of IP vs input. ChIP-seq of H3K27me3 is shown as mean signal of two replicates at each base

---

copies. Therefore, most $2 \times 150$ bp reads can be mapped uniquely and assigned to individual L1 loci, with the possible exception of reads originating from a few young L1s that display particularly low mapability and polymorphic L1 alleles that are not in the hg38 reference genome.

The increased FLI-L1 transcription was only detected in the sense orientation, indicating that the L1 promoter is driving the expression of these elements (Fig. 2f). Using in-depth proteomic profiling and western blot analysis, we detected a massive increase in levels of L1 ORF1 peptides (ORF1p) in *DNMT1*-KO cells at the same time as the transcriptional activation of FLI-L1s (Fig. 2h, Supplementary Fig. 1p). This demonstrates that loss of DNA methylation results in FLI-L1s that are expressed at high levels, compatible with substantial protein translation.

**DNA demethylation leads to chromatin remodeling of young L1s.** Despite the substantial loss of CpG methylation upon LV. CRISPR-DNMT1 transduction, we did not detect any major transcriptional activation of any classes of TEs other than young L1s or LTR12C elements. To obtain further insight into the mechanism underlying the specific upregulation of evolutionarily young L1s, we looked in detail at the level of individual elements in both recent and ancient full-length L1s (>6 kb). We also investigated full-length human endogenous retrovirus (HERV) and SINE-VNTR-Alu (SVA) elements, which both represent young TEs with documented transcriptional activity in certain human cell types. Analysis of DNA methylation levels using WGBS data demonstrated that virtually all L1s, HERVs, and SVAs were extensively methylated in hNPCs, and that this DNA methylation was lost in *DNMT1*-KO cells (Fig. 2i, Supplementary Fig. 3a). As noted above, the loss of DNA methylation resulted in the expression of TE-derived RNA from younger L1s (L1HS, L1PA2, and L1PA3) while older L1s, HERVs, and SVAs were not expressed, even in the absence of DNA methylation (Fig. 2i).

To confirm that L1-derived transcripts were detected because of transcriptional activation, we performed ChIP-seq experiments for RNA polymerase II (Pol II). This analysis confirmed that the transcriptional machinery is recruited specifically to young L1s upon loss of DNA methylation, but not to old L1s, HERVs, or SVAs (Fig. 2i, Supplementary Fig. 3a). To investigate if L1 activation was associated with chromatin remodeling, we performed ChIP-seq for the histone mark H3K27ac, which is associated with transcriptional activation, and found that the loss of DNA methylation resulted in the acquisition of H3K27ac at the 5' end of young L1s but not in other TEs (Fig. 2i, Supplementary Fig. 3a). It is worth noting that we detected chromatin remodeling and activated transcription from virtually all young L1s,

demonstrating a group-like behavior, and suggesting that the integration site of the L1 does not influence transcriptional activation following loss of DNA methylation. Together, these results demonstrated that while both young and old L1s, as well as other TEs, are DNA-methylated, loss of DNA methylation only results in chromatin remodeling and transcriptional activation in evolutionarily young L1s.

These observations suggested that young L1s have unique properties that enable their transcriptional activation upon loss of DNA methylation in hNPCs. One possibility is that the TEs which do not show any transcriptional activation upon DNA methylation loss are controlled by additional repressive factors. Therefore, we analyzed ChIP-seq data for the presence of the repressive chromatin marks H3K9me3 and H3K27me3, which have both been implicated in the control of L1s and other TEs. Surprisingly, we found that all young and old L1s, SVAs, and HERVs carry H3K9me3 but not H3K27me3 in hNPCs (Fig. 2j, Supplementary Fig. 3b). Depositing H3K9me3 on TEs is known to depend on the epigenetic co-repressor protein TRIM28 that is recruited to TEs via sequence-specific binding of KRAB-zinc finger proteins (KRAB-ZFPs)[27]. Accordingly, we found that TRIM28 is also bound to all young and old L1s, SVAs, and HERVs in hNPCs (Fig. 2j, Supplementary Fig. 3b). These data demonstrate that a dual-repressive layer of DNA methylation and TRIM28/H3K9me3 covers all young TEs in hNPCs. The fact that loss of DNA methylation results in the specific transcriptional activation of young L1s suggests that these elements are capable of overriding the negative TRIM28/H3K9me3 signal through an as yet unidentified mechanism.

**Activated primate-specific L1s act as alternative promoters.** Since demethylated young L1s acquired the histone mark H3K27ac, associated with active promoters and enhancers, we hypothesized that they have the capacity to act as gene regulatory elements and influence the expression of nearby protein-coding genes. Indeed, we found that protein-coding genes located within 50 kb of activated L1HS, L1PA2, or L1PA3 were significantly upregulated in *DNMT1*-KO hNPCs, while genes located close to HERVs and SVAs were not (Fig. 3a, b).

To investigate the underlying mechanism of nearby activated genes, we analyzed DNA methylation levels and RNA levels, as well as the presence of H3K27ac and Pol II at these genes and in the surrounding genome, including the adjacent L1 elements. We found that the majority of these genes already had a demethylated promoter in the control setting, as well as displaying both H3K27ac and Pol II at the transcriptional start site (TSS) (Fig. 3c). Removal of DNA methylation resulted in upregulation of RNA

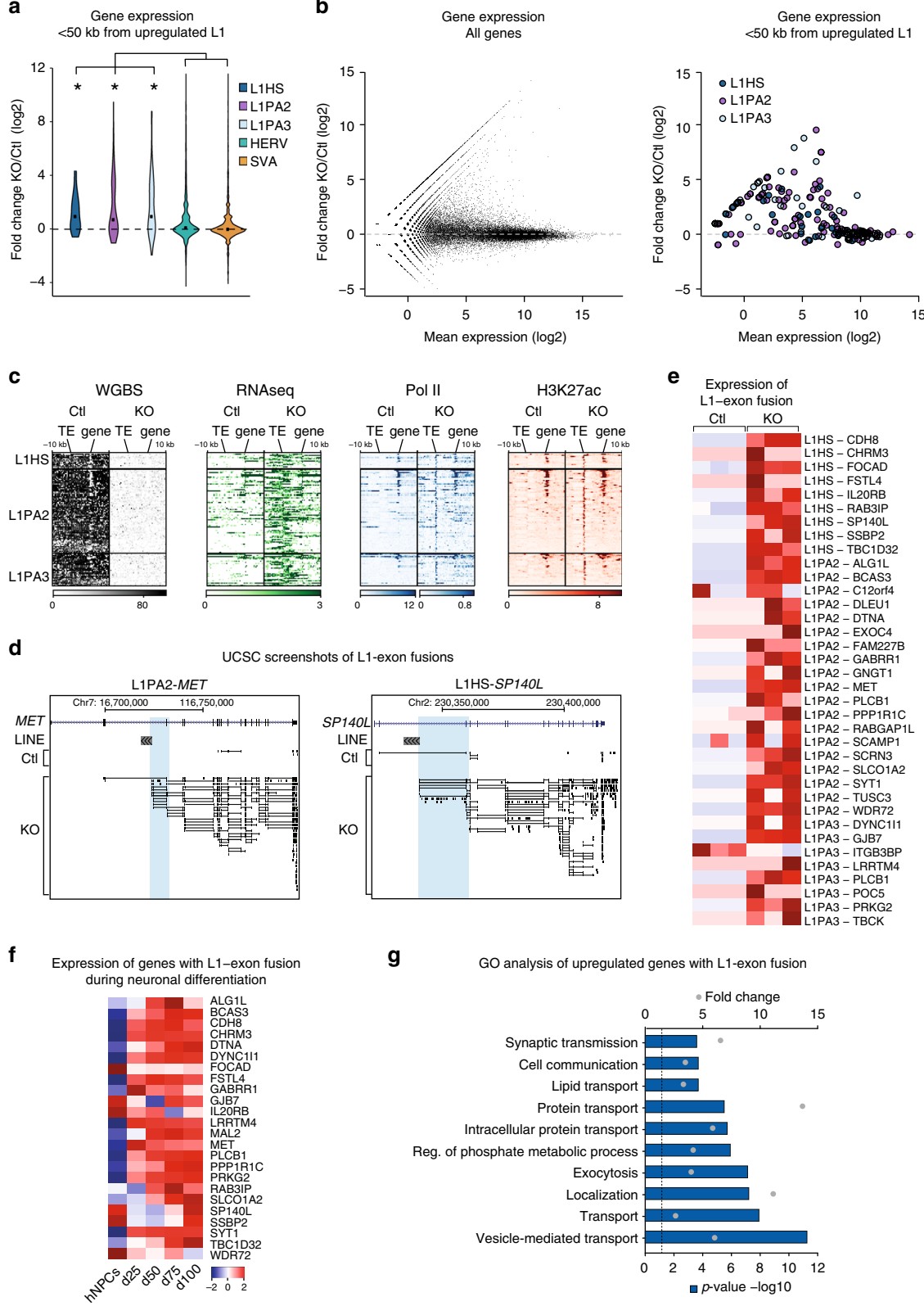

**a** Gene expression <50 kb from upregulated L1

**b** Gene expression All genes

Gene expression <50 kb from upregulated L1

**c** WGBS | RNAseq | Pol II | H3K27ac

**d** UCSC screenshots of L1-exon fusions

L1PA2-*MET*

L1HS-*SP140L*

**e** Expression of L1−exon fusion

**f** Expression of genes with L1−exon fusion during neuronal differentiation

**g** GO analysis of upregulated genes with L1-exon fusion

produced from the loci, despite no apparent change in H3K27ac and Pol II at the TSS of the gene. We rather found that recruitment of H3K27ac and Pol II was limited to the nearby L1 element, suggesting that the increase in transcription originates from the TE (Fig. 3c).

Since L1s can act as alternative promoters for endogenous protein-coding genes[28,29], we thought that such a phenomenon may underlie the effects we see on nearby gene expression. Thus, we searched for incidences where L1s act as alternative promoters upon *DNMT1*-KO. We first identified TSS coordinates of protein-coding genes in gencode (v25) that overlap L1 elements and found 131 primate-specific L1s with potential alternative promoter activity (Supplementary Data 2). In line with previous reports[28,29], the majority of these elements were in the antisense

**Fig. 3** Demethylated L1 elements act as alternative promotors and drive the expression of neuronal genes. **a** The expression of genes within 50 kb of upregulated L1s, FL-HERVs, and SVAs upon *DNMT1*-KO (*n* = 3 in both groups), *p* < 0.05, Wilcoxon rank sum test. The black dot shows the mean expression. **b** The expression of genes that lie within 50 kb of upregulated L1-HS, L1-PA2, L1-PA3. **c** Heatmaps of WGBS, RNA-seq, ChIP-seq of Pol II and H3K27ac in control and *DNMT1*-KO hNPCs of genes that lie within 50 kb of upregulated L1-HS, L1-PA2 and L1-PA3. **d** UCSC screenshots of two examples of L1-fusion transcripts. Blue boxes are indicating the reads that are initiated in the L1s. **e** A heatmap showing the relative expression of L1-exon fusion reads in three biological replicates of *DNMT1*-KO and control hNPCs. Only L1-exon fusion reads for genes in which the L1 overlap annotated TSS are included. **f** A heatmap showing the expression of L1-driven genes upon neuronal differentiation of hNPCs (*n* = 2 at all timepoints). Only genes that are significantly upregulated in *DNMT1*-KO hNPCs and have a hominoid-specific L1-TSS overlap as well as L1-exon fusion reads in the neuronal differentiation data are included. **g** A GO analysis of the L1-driven genes, using Panther BP ontologies, *p* < 0.05 (*p* = 0.05 is indicated with a dotted line). The top 10 terms are listed, for a full list see Supplementary Data 3

orientation relative to the gene. We found that many of these L1 promoters were activated upon *DNMT1*-KO, as identified by fusion reads between the L1 and an exon (Fig. 3d, e). RT-PCR verification confirmed the existence of the L1-gene fusions (Supplementary Fig. 4a). The same L1-fusion transcripts were found upon targeting *DNMT1* in the additional iPSC-derived hNPC line (Supplementary Fig. 4b). In addition, we noted that some of the activated LTR12C elements also gave rise to fusion transcripts (Supplementary Fig. 4c).

Primate-specific L1s contain a short open reading frame in the antisense orientation (ORF0) that can be spliced into exons of nearby genes resulting in fusion proteins[30]. Since several of the L1-fusion exons that we detected have been previously demonstrated to be such ORF0-gene fusion transcripts[30], we investigated the expression of these genes at the protein level using the in-depth proteomics data. We observed increased protein levels for several of L1-fusion genes, suggesting that the L1-mediated influence on transcriptional levels is, at least in some instances, directly related to changes in protein level and does not simply represent transcriptional noise (Supplementary Fig. 4d).

We next sought to functionally characterize the genes influenced by L1 activation: we found that most were upregulated upon neuronal differentiation of hNPCs (Fig. 3f). In line with this, GO analysis confirmed that these genes were enriched for functions such as synaptic transmission and cell communication (Fig. 3g, Supplementary Data 3). Interestingly, several of the L1-regulated genes have previously been implicated in neurodevelopmental disorders such as autism (*CDH8*, *GABRR1*, *MET*)[31–33], cognitive impairment (*SYT1*, *TUSC3*, *TBCK*)[34,35], and epilepsy (*PLCB1*)[36]. To confirm that the observation of L1-fusion transcripts was not an artefact of the cell culture system, we performed long-read strand-specific polyA-enriched RNA-seq of sub-dissected samples from different regions from the developing human brain, including both proliferative as well as more mature stages of neuronal differentiation, and found that several L1-fusion transcripts could be detected. (Supplementary Fig. 4e).

Taken together, these data demonstrate that activation of primate-specific L1s as a consequence of DNA demethylation results in L1-driven transcriptional enhancement of many protein-coding genes involved in neuronal functions and psychiatric disorders. This implicates young L1s in the control of transcriptional networks in neural cells, making an additional dimension to how these elements influence brain function.

## Discussion

It has long been thought that DNA methylation is the key determinant of TE silencing in somatic cells. However, the actual impact of DNA methylation in somatic cells has proven difficult to study, mainly due to the fact that loss of DNMT1 in mouse models and human cancer cells leads to cell death[19,20]. In this study, we deleted *DNMT1* using CRISPR-Cas9 in hNPCs, and found that these cells survived and remained proliferative despite

a global loss of DNA methylation. These results highlight that the downstream consequences of demethylation in human and mouse cells are fundamentally different, as has already been noted in embryonic stem cells[22].

Interestingly, we found that only evolutionarily young L1s (those found exclusively in hominoids) are activated by loss of DNA methylation, while the majority of TEs remain transcriptionally silent. This includes older L1 integrants that are shared between primates and other mammals, as well as both ancient and primate-specific TEs of other classes. While some of these observations can be explained by the higher burden of mutations and genomic rearrangements of ancient TEs that reduce their capacity to be transcribed, it is worth to note that many evolutionarily young elements, such as full-length HERVs and SVAs also remained silent upon DNA demethylation. We found that most young TEs carry a dual layer of repression, characterized by the repressive histone mark H3K9me3 in addition to DNA methylation. In line with this, a number of recent studies have demonstrated that the epigenetic co-repressor protein TRIM28 is recruited to TEs in somatic cells via the binding of KRAB-zinc finger proteins (KRAB-ZFPs) and thereby establish H3K9me3 marks[11,12,37–39].

Why only young L1s are able to escape this dual-layer repression by removing DNA methylation remains unclear. However, it is known that the evolutionary age of L1s influence how they are silenced in pluripotent stem cells, where young L1s are controlled by PIWI-associated mechanisms while older L1s are silenced by TRIM28/H3K9me3[40,41]. This difference may also reflect the dependency of DNA methylation in somatic cells. Alternatively, it is possible that hNPCs express transcription factors that provide a robust activation of L1s even in the presence of H3K9me3, while such transcription factors necessary for e.g. HERV and SVA expression are lacking in this cell type[42] or that TRIM28 is recruited to young L1s through a DNA methylation-dependent KRAB-ZFP[43].

Our data suggest that L1s that escape DNA methylation will be highly transcribed in the brain and can then potentially retrotranspose into a new position in the host genome, as has been demonstrated by several independent studies[5–8,10]. However, due to the random nature of the transposition it has been difficult to link these events to physiological impact. We propose that the presence of L1s scattered throughout the mammalian genome provides a potential source of widespread gene regulatory networks. L1 activation leads to a network-like activation of genes influenced by alternative promoter activity of nearby young L1 elements, thereby influencing the expression level of many protein-coding genes linked to neuronal functions and psychiatric disorders. Indeed, alterations in DNA methylation has been linked to several neurodevelopmental disorders such as autism and schizophrenia[44,45], and mutations in *DNMT1* lead to a broad spectrum of neurological disorders that include cognitive disturbances and psychiatric manifestations[46]. With this in mind, it is interesting to note that many of the genes where L1 acts as an

alternative promoter upon loss of DNA methylation have been directly implicated in neurodevelopmental disorders. Thus, loss of DNA methylation at young L1s upstream of such genes would directly influence their expression. This raises the hypothesis that alterations in DNA methylation patterns of L1s during neurodevelopment directly impact on genes implicated in neurodevelopmental disorders. Since L1s are highly polymorphic within the human population, the prevalence of certain L1 copies in the genome could thereby directly influence the etiology of neurodevelopmental disorders.

## Methods

**Cell culture**. The DNMT1-KO was performed on the embryo-derived human neural epithelial-like stem cell line Sai2[23], and was maintained according to standard protocol[47]. The DNMT1-KO results were verified in an iPS-derived human neuroepithelial-like stem cell line AF22[47].

**Lentiviral vectors and transduction**. LV.gRNA.CAS9-GFP vectors[22] were used to target DNMT1 (LV.CRISPR-DNMT1) or LacZ (control). Lentiviral vectors were produced according to Zufferey et al.[48] and were in titers of $10^9$ TU/ml which was determined using qRT-PCR[49]. hNPCs were transduced with a MOI of 10–15 and allowed to expand 10 days prior to FACS (FACSAria, BD sciences). Cells were detached and resuspended in basic culture media with Rock inhibitor (10 μM, Miltenyi) and propidium iodide (BD Bioscience), and strained (70 μm, BD Bioscience). GFP-positive cells were spun down (400 g, 7 min) and snap frozen on dry ice. Cell pellets were kept at −80 °C until RNA/DNA was isolated. All groups were performed in biological triplicates.

**Immunocytochemistry and BrdU quantification**. Cells were fixed in 4% paraformaldehyde for 15 min and incubated overnight with the primary antibody (5mC, Active Motif, cat.no. 39649, lot 06116002, used 1:250; NESTIN, BD, cat.no 611658, lot 45333, used 1:500; SOX2, R&D, cat.no MAB2018, lot KGQ0200041, used 1:50; BrdU, Serotech, cat.no OBT0030, lot 0512, used 1:250), followed by a 2 h incubation with a fluorophore-conjugated secondary antibody (Jackson Laboratories) and DAPI. Cells stained for 5mC and BrdU were pre-treated with 0.9% Triton in PBS for 15 min followed by 2 N HCl for 15 min followed by 10 mM Tris-HCl, pH 8, for 10 min prior to incubation with the primary antibody. Imaging of cells was performed with a fluorescence microscope (Leica).

BrdU was incorporated by adding BrdU (10 μM, Life Technologies) to the culture media of different wells for 5 consecutive days, starting on day 10 after FACS, and fixing the cells 24 h after exposure. Cells were stained for Dapi, BrdU, and 5mC. The 5mC expression and BrdU incorporation was evaluated using a Cellomics Scan 6.6.0 from Thermo Scientific and HCS Studio. Cells were quantified as Dapi + /BrdU + /5mC + in the control group and Dapi + /BrdU + /5mC− in the DNMT1-KO group. On average 11,906 control cells and 7,774 DNMT1-KO cells were analyzed per time point.

**Analysis of CRISPR-Cas9 mediated indels**. Total genomic DNA was isolated from control and DNMT1-KO hNPCs, and the sequence of the target site was amplified by PCR (a 1.4 kb fragment centering on the target site) and subjected to NexteraXT fragmentation, according to manufacturer recommendations. Indexed tagmentation libraries were sequenced with 2 × 150 bp PE reads and analyzed using an in-house TIGERq pipeline to evaluate CRISPR-Cas9 editing efficiency. Here, reads carrying indels across the target site were assembled and normalized to define a list of unique indels presented in the tested sample, with their relative frequencies. Lastly, indels were further subset into frame-shifting and frame-maintaining variants to estimate indel diversity. Primers are found in Supplementary Data 4.

**RNA analysis**. Total RNA from hNPCs and human embryonic tissue was isolated using the RNeasy Mini Kit (Qiagen) and used for RNAseq and qPCR. Human tissue was obtained from legally aborted embryos from post-conception week 6.5–11, with approval of the Swedish National Board of Health and Welfare.

Libraries from DNMT1-KO and control hNPCs were generated using Illumina TruSeq Stranded mRNA library prep kit (poly-A selection) and sequenced on a HiSeq2500 (PE 2 × 150 bp). For all RNAseq data sets, the sequencing reads were mapped to the human reference genome (hg38) using STAR aligner v2.5.0a[50] and quantified using the Subread package FeatureCounts[51]. To quantify NCBI gene[52] entries, reads mapping to 10 or fewer positions were kept, and the primary alignment were used for quantification. Mapping was performed with a splice junction overhang of minimum one base, assisted by Gencode v25 annotation.

To analyze retrotransposons, we discarded all reads mapping equally well to more than one locus (–outFilterMultimapNmax 1). If a read aligned to multiple positions, but one alignment had a better alignment score than any other, it was included (–outFilterMultimapScoreRange 1). A maximum of 0.03 mismatches per base were allowed (–outFilterMismatchNoverLmax 0.03). RepeatMasker annotation for hg38 was downloaded from the UCSC table browser[53].

Mapping reads to very recent L1 integrants is challenging, as their high sequence similarity causes reduced mapability. However, the average divergence of L1-HS integrants from consensus sequence was 1.37% (RepeatMasker), meaning on average about one mutation per 80 bases. Thus, at least some fragments of most L1-HSs will be mappable with paired-end 2 × 150 bp strand-specific sequencing. However, it is worth noting that some activated young L1s might not be detected in our analysis due to low mapability and low abundance. In addition, our approach will not detect potential expression of polymorphic L1 alleles that are not in the hg38 reference genome.

Differential expression analysis of DNMT1-KO data was performed using the R package DESeq2[54]. RefSeq and RepeatMasker counts were analyzed separately. RefSeq counts were normalized using the default median ratio method of DESeq2. All transposons with fewer than five reads in total across all samples were filtered out. Transposon counts were normalized sample-wise by scaling to the number of reads mapping to the genome.

cDNA for qRT-PCR was created with the Maxima First Strand cDNA Synthesis Kit (Invitrogen) and analyzed with SYBR Green I master (Roche) on a LightCycler 480 (Roche). Data are represented with the ΔΔCt method normalized to the housekeeping genes GAPDH and HPRT1. Error bars represent SEM from three biological and three technical replicates. Primers are listed in Supplementary Data 4.

**Whole-genome bisulfite sequencing data**. 10 days post-transduction, 200,000 sorted cells of both control and DNMT1-KO hNPCs were treated with EpiTect Bisulphite kit (Qiagen) kit according to manufacturer's instructions and WGBS libraries were generated and sequenced according to Liao et al [22].

The raw 2 × 150 bp reads were trimmed for adapter sequences and low-quality reads with trimmomatic[55] using a sliding window of four bases, with quality cut-off at 20. The 20 leading and trailing bases were trimmed if below 20. Reads with fewer than 50 bases after trimming were discarded. Furthermore, the first 20 bases on read mate 2 were cropped using cutadapt due to overrepresentation of guanine bases. Fastqc was used to assess quality.

The trimmed and filtered reads were further processed with Bismark[56]. Bisulfite genome conversion and indexing was done with bowtie2[57] running bismark_genome_preparation, before running Bismark mapping, deduplication, and methylation extraction, using–no_overlap to avoid coverage bias. The output bedGraph coverage files were converted to bigwig using the UCSC tool bedGraphToBigWig[58] for visualization and analysis.

To calculate CpG methylation at TEs, the BEDOPS (v2.4.35,[59]) bedmaps tool was used to calculate mean mCpG/CpG ratio over TE fragments. Only CpG sites with at least 3 × coverage were included. The number of CpG sites included in analysis was counted using bedmaps –count.

**Western blot**. 200,000 GFP + cells were sorted from both Control and DNMT1-KO cells and lysed in RIPA buffer (Sigma-Aldrich) containing Protease inhibitor cocktail (PIC, Complete, 1:25) and kept on ice for 30 min before spun at 10,000×g for 10 min at 4 °C. Supernatants were collected and transferred to a new tube and stored at −20 °C. Each sample was mixed 1:1 (10 μl + 10 μl) with Laemmli buffer (BioRad) and boiled at 95 °C for 5 min before loaded onto a 4–12% SDS/PAGE gel and run at 200 V. Gel was electrotransferred using Transblot-Turbo Transfer system (BioRad). The membrane was then washed 2 × 15 min in TBS with 0.1% Tween20 (TBST) and blocked for 1 h in TBST with 5% nonfat dry milk before incubating the membrane at 4 °C overnight with the mouse anti-L1 ORF1p (Millipore Cat MABC1152, Lot 3137651, 1:1000) diluted in TBST with 5 % non-fat dry milk. The membrane was washed in TBST 2 × 15 min and incubated for 1 h in room temperature with HRP-conjugated anti-mouse antibody (Santa Cruz Biotechnology cat.no sc-516102, lot. B0217, 1:3000) diluted in TBST with 5% non-fat dry milk. After washing the membrane 2 × 15 min in TBST and 1 × 15 min in TBS, the protein expression was revealed by chemiluminescence using Immobilon Western (Millipore) and the signal detected using a Chemidoc MP system (BioRad). The membrane was stripped by treating it with methanol for 15 s followed 15 min in TBST before incubating it in stripping buffer (100 mM 2-mercaptoethanol, 2% (w/v) SDS, 62.4 mM Tris-HCL pH 6.8) for 30 min 50 °C. The membrane was washed in flowing water for 15 min followed by 3 × 15 min in TBST before blocked for 1 h in TBST with 5% nonfat dry milk. The procedure for the β-actin staining (mouse anti-β-actin HRP, Sigma-Aldrich cat.no A3854, lot. 034M4830V 1:50,000) was then performed as above.

**Analysis of L1-gene fusion transcripts**. To identify L1 and LTR12C promoters, all TSSs for transcripts of protein-coding genes annotated in Gencode v25[60] were intersected with L1 or LTR12C coordinates using the BEDTools (v2.26.0) intersect module[61]. To detect fusion reads between TEs and genes, the aligned bam files were scanned for reads that overlapped TEs. These reads were further scanned for overlap with transcripts of Gencode protein-coding genes. For protein-coding genes, we only included reads mapping in sense direction relative to the protein-coding gene, while reads mapping in both sense and antisense to the L1 or LTR12C were kept, allowing detection of both sense and antisense L1 and LTR12C promoters.

**ChIP-sequencing**. Crosslinking of the hNPCs (*DNMT1*-KO: 140,000 cells, Control: 400,000 cells) was performed 10 days post transduction by resuspending cells in 0.5 ml PBS containing 2% FCS, and adding an equal volume of 2% PFA (Thermofisher, no. 28908). Tubes were mixed well and rotated in darkness for 10 min. 0.1 ml of 1 M glycine in PBS was added, and tubes were mixed well before rotating in darkness for 10 min. Tubes were spun at 2,000 g for 10 min at 4 °C, followed by a wash with 0.1 M glycine in PBS and spun again before storing at −80 °C. Chromatin immunoprecipitation and tagmentation was carried out as described by Gustafsson et al.[62] Per ChIP, 3 µg of anti-H3K27Ac (Diagenode, cat.no C15410196, lot A1723–0041D), anti-Pol II (RPB2, N-term; GeneTex, cat.no GTX102535-S, lot 39918) or anti-H3K27me3 (Millipore; cat.no 07–449, lot 3018864), was added to 10 µl protein G-coupled Dynabeads (ThermoFisher) in PBS with 0.5% BSA and incubated with rotation for 4 h at 4 °C.

The crosslinked cells were thawed at room temperature, pelleted, and diluted with SDS lysis buffer (50 mM Tris/HCl pH8, 0.5% SDS and 10 mM EDTA pH8) and placed cold for 15 min. Cells were sonicated for 12 cycles of 30 s on/30 s off on high power using a Bioruptor Plus (Diagenode). To neutralize the SDS, Triton X-100 was added to a final concentration of 1% along with 2 µl 50 × cOmplete protease inhibitor (final concentration 1x). Samples were incubated at room temperature for 10 min. Dynabeads coated with anti-H3K27Ac or anti-Pol II antibodies were washed with PBS containing 0.5% FCS, and mixed with cell lysate in PCR tubes. Tubes were incubated overnight at 4 °C. For H3K27me3 immunoprecipitation, cell lysate supernatants from Pol II ChIP were collected after overnight incubation, directly merged with pre-washed dynabeads coated with anti-H3K27me3 antibodies and subjected to another overnight incubation.

Immunoprecipitated chromatin was washed with 150 µl of low-salt buffer (50 mM Tris/HCl, 150 mM NaCl, 0.1% SDS, 0.1% NaDOC, 1% Triton X-100, and 1 mM EDTA), high-salt buffer (50 mM Tris/HCl, 500 mM NaCl, 0.1% SDS, 0.1% NaDoc, 1% Triton X-100, and 1 mM EDTA) and LiCl buffer (10 mM Tris/HCl, 250 mM LiCl, 0.5% IGEPAL CA-630, 0.5% NaDOC, and 1 mM EDTA), followed by two washes with TE buffer (10 mM Tris/HCl and 1 mM EDTA) and two washes with ice-cold Tris/HCl pH8. For tagmentation, bead-bound chromatin was resuspended in 30 µl of tagmentation buffer, 1 µl of transposase (Nextera, Illumina) was added and samples were incubated at 37 °C for 10 min followed by two washes with low-salt buffer. Bead-bound tagmented chromatin was diluted in 30 µl of water. 15 µl PCR master mix (Nextera, Illumina) and 5 µl indexed amplification primers (ref Buenrostro ATACseq) (0.125 µM final concentration) were added and libraries prepared using the following program: 72 °C 5 min (adapter extension); 95 °C 5 min (reverse cross-linking); followed by 11 cycles of 98 °C 10 s, 63 °C 30 s, and 72 °C 3 min.

After PCR amplification, library cleanup was done using Agencourt AmPureXP beads (Beckman Coulter) at a ratio of 1:1. DNA concentrations in purified samples were measured using the Qubit dsDNA HS Kit (Invitrogen). Libraries were pooled and single-end sequenced (50 cycles) using the Nextseq500 platform (Illumina). The raw 50 bp fastq files were trimmed and filtered with trimmomatic, and quality was assessed with fastqc. The reads were then mapped to hg38 with bowtie2 with –sensitive-local mapping parameters. We next filtered out multi-mapping reads but removing all reads with MAPQ < 10 using Samtools[63], keeping only reads with one unique best alignment. Bam files were converted to bedGraph and bigwig using UCSC tools. Deeptools[64] were used for quality control, analysis, and visualization of ChIP-signals at various genomic features. Coordinates for transposable elements were downloaded from UCSC table browser[53]. For visualization, the mean signal of two replicates was calculated for each treatment (control and *DNMT1*-KO) for each ChIP (H3K27ac, H3K27me3, and Pol II) using the bamCompare tool from Deeptools. For TRIM28 and H3K9me3, only one ChIP reaction was used; signals from these were normalized to log2 ratio between IP and input samples using the Deeptools bamCompare module. As a positive control for the H3K27me3 ChIP, we found strong H3K27me3 signals at the HOX gene cluster (Supplementary Fig. 5).

**GO term analysis**. Gene ontology enrichment analysis was performed using the Panther Overrepresentation test (release 20170413) with Panther version 12.0 released 10–07–2017. To evaluate significance, all genes with at least 3 reads on average in control samples were used as the reference background list. Enrichment of the Panther GO biological process and Panther pathway terms were calculated separately.

**Proteomic analysis**. Cells were sorted on day 10 and frozen at −80 °C. Following cell lysis, proteins were reduced/alkylated and digested with trypsin (Promega). Protein digests were differentially dimethyl labeled on column[65]. Briefly, samples were labeled by flushing the columns with labeling reagent (light or heavy using $CH_2O + NaBH_3CN$ or $CD_2O + NaBH_3CN$ (all Sigma-Aldrich), respectively). Sample complexity was reduced by fractionation using OFFGEL isoelectric focusing (Agilent). The 12 fractions resolved were acidified and desalted with C18 Ultra-Micro SpinColumns (Harvard). Peptide samples were dried by vacuum centrifugation and stored at −20 °C until further use. Samples were reconstituted in 4% acetonitrile/0.1% formic acid prior to MS analysis.

MS analyses were carried out on an Orbitrap Fusion Tribrid MS system (Thermo Scientific) equipped with a Proxeon Easy-nLC 1000 (Thermo Fisher). Injected peptides were trapped on an Acclaim PepMap C18 column (3 µm particle size, 75 µm inner diameter × 20 mm length, nanoViper fitting). After trapping, gradient elution of peptides was performed on an Acclaim PepMap C18 100 Å column (3 µm particle size, 75 µm inner diameter × 150 mm length, nanoViper

fitting). The mobile phases for LC separation were 0.1% (v/v) formic acid in LC-MS grade water (solvent A) and 0.1% (v/v) formic acid in acetonitrile (solvent B). Peptides were first loaded with a constant flow of solvent A at 9 µl/min onto the trapping column. Subsequently, peptides were eluted via the analytical column at a constant flow of 600 nl/min. During the elution step, the percentage of solvent B increased in a linear fashion from 5% to 10% in 2 min, then increased to 25% in 85 min, and finally to 60% in a further 20 min. The peptides were introduced into the mass spectrometer via a stainless steel nano-bore emitter 150 µm outer diameter × 30 µm inner diameter; 40 mm length (Thermo Fisher Scientific) and a spray voltage of 2.0 kV was applied. The capillary temperature was set at 275 °C.

One full-scan spectrum from m/z 375 to 1500 at a resolution of 60,000 FWHM was followed by MS/MS scans (resolution 15,000 FWHM) of the most intense ions (up to 15) from the full-scan MS. The precursor ions were isolated with 1.6 m/z isolation width and fragmented using higher-energy collisional-induced dissociation at a normalized collision energy of 30%. The dynamic exclusion time was set to 30 s. The automatic gain control was set to 4e5 and 5e4 for MS and MS/MS, respectively, and ion accumulation times of 50 ms (MS) and 60 ms (MS/MS). The intensity threshold for precursor ion selection was 5e4.

MS raw data files were processed with MaxQuant (version 1.5.0.0[66]. Enzyme specificity was set to trypsin/P and a maximum of two missed cleavages were allowed. Cysteine carbamidomethylation and methionine oxidation were selected as fixed and variable modifications, respectively. The derived peak list was searched using the in-built Andromeda search engine in MaxQuant against the Uniprot human database together with the Cas9 and GFP genes used in this study, as well as 265 frequently-observed contaminants (Andromeda configured database) and reversed sequences of all entries. Initial maximal allowed mass tolerance was set to 20 ppm for peptide masses, followed by 6 ppm in the main search, and 20 ppm for fragment ion masses. The minimum peptide length was set to seven amino acid residues. A 1% false discovery rate (FDR) was required at both the protein level and the peptide level. In addition to the FDR threshold, proteins were considered identified if they had at least one unique peptide. The protein identification was reported as an indistinguishable "protein group" if no unique peptide sequence to a single database entry was identified. The "match between runs" was enabled for consecutive peptide fractions with a 1.5 min time window. Ratio count 1 was used. Contaminants, reverse hits and "hits-only-identified-by-site" were excluded. The iBAQ algorithm was used to estimate the abundance of different proteins within a single sample[67].

**Reporting summary**. Further information on research design is available in the Nature Research Reporting Summary linked to this article.

## Data availability
There are no restrictions in data availability. RNA and DNA sequencing data presented in this study have been deposited at Gene Expression Omnibus with the accession code GSE107580. Mass spectrometry proteome data have been deposited at the ProteomeXchange Consortium via the Proteomics Identifications Database (PRIDE) partner repository[68] with the dataset identifier PXD008648. The source data underlying Fig. 2h, Supplementary Figs. 1b–e, k, n–o are provided as a source data file.

## Code availability
All codes can be retrieved by contacting the corresponding author.

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

## Acknowledgements

We would like to thank D. Trono and C. Bellodi for helpful comments on the manuscript and A. Falk for providing the hNPC lines. We are grateful to all members of the Jakobsson lab. We also thank J. Johansson, M. Persson Vejgården, U. Jarl, and A. Hammarberg for technical assistance. The work was supported by grants from the Swedish Research Council, the Swedish Foundation for Strategic Research, the Swedish Brain Foundation, the Swedish excellence project Basal Ganglia Disorders Linnaeus Consortium (Bagadilico), and the Swedish Government Initiative for Strategic Research Areas (MultiPark & StemTherapy). Support from the Swedish National Infrastructure for Biological Mass Spectrometry (BioMS) is gratefully acknowledged.

## Author contributions

All authors took part in designing the study as well as interpreting the data. M.E.J and J.J. conceived and designed the study. M.E.J., R.P., C.G., K.P, S.V., S.M., D.Y. and J.H.

performed experimental research and P.L.B. performed bioinformatical analyses. J.L., R.M., and A.M. contributed expertise and reagents. M.E.J., P.L.B. and J.J. wrote the paper and all authors reviewed the paper.

## Additional information

**Competing interests:** The authors declare no competing interests.

