## [Peer Review File · Nature Communications]

Reviewers' comments:

Reviewer #4 (Remarks to the Author):

This is an interesting paper describing retrotransposon derepression after CRISPR-Cas mediated mutagenesis of DNMT1 DNA methyltransferase in two independent neural progenitor lines. The authors report that evolutionary young LINE elements (and some other types of evolutionary young retrotransposon) are particularly affected by hyperexpression/de-repression after loss of DNA methylation. This affects also the expression of disease-relevant protein-coding genes that are in proximity of LINE1 elements, including numerous neurodevelopmental risk genes.

The novelty of this paper is two-fold (i) the phenotype is milder than in DNMT1 mouse 'knock-out' cells and (ii) the massive loss of DNA methylation after DNMT1 loss of function mutagenesis primarily derepresses the evolutionary younger type of repeat elements.

The paper appears to be improved since initial submission to Nature journal series, for example the additional data shown in figure 1d-f, documenting massive loss of methylation by WGBS method provide convincing evidence that the mutant neural progenitor cells indeed are affected by massive DNA methylation loss.

Major comments:

1. The most exciting question in the field (does derepression of young LINE1 elements increase the number of de novo insertion sites with somatic mutagenesis in these neural progenitors) is not addressed in this paper.
2. Because evolutionary older repeat elements have a much higher burden of degenerative mutations, rearrangement etc., isn't it expected that an epigenetic alteration such as loss of DNA methylation primarily results in derepression of the younger type of repeat elements. This is not properly discussed, I feel.
3. Some important comments raised in previous review, including the recommendation to confirm the acclaimed 'massive increase in levels of L1ORF protein' by western blot, are still unaddressed.

Reviewer #5 (Remarks to the Author):

Jönsson et al have generated DNMT1 KO hNPC cells, which surprisingly remain viable, and found pronounced upregulation of young L1 elements, some of which generate chimeric transcripts with neuronal genes.

These are key findings when it comes to understanding the role of DNA methylation and other repressive mechanisms in TE (and gene) regulation in differentiated tissues, about which we know so little. The data backing the dependency of young L1 repression on DNMT1 are extensive and robust, which includes a careful navigation of the caveats associated with mapping of short read data to TEs. The observations on chimeric transcripts are also of interest, although the link between L1 demethylation and gene activation in neuronal differentiation and/or neurodevelopmental disorders is unclear.

I have the following suggestions for the authors' consideration:

- 1) Is there evidence from RNA-seq data that the L1 fusion transcripts identified here are expressed upon hNPCs differentiation and/or in neurodevelopmental disorders? This would provide a stronger link with a physiological context.

2) Although L1s are the most prominently affected TEs in the KO, some LTRs are also greatly upregulated. Notably, many copies of LTR12C elements are derepressed, and these have also been previously linked with the generation of chimeric transcripts (e.g., PMID: 28604729). It would be interesting to know to what extent this LTR family is also generating chimeric transcripts in hNPCs and, if so, how they compare with the ones from L1s with regards to gene ontology and expression patterns.

3) The authors acknowledge in the Methods section that polymorphic L1s can affect the outcome of sequencing data mapping. This should probably be mentioned in the main text, as it is an important point.

The following are non-essential points that mainly stem from my curiosity about the data:

4) I cannot help wonder what happens to H3K9me3/TRIM28 at different L1 families in the DNMT1 KO cells. It would be great if the authors had some ChIP-qPCR data on this.

5) I take it from the GO analysis in Figure S1h (and from cell survival) that L1 activation does not lead to an interferon response, as seen elsewhere. This may be worth mentioning.

6) Do the authors have any data on whether the piRNA pathway becomes active upon DNMT1 KO?

7) Do DNMT1 KO hNPCs die upon differentiation?

Miguel Branco

Reviewers' comments:

Reviewer #4 (Remarks to the Author):

This is an interesting paper describing retrotransposon derepression after CRISPR-Cas mediated mutagenesis of DNMT1 DNA methyltransferase in two independent neural progenitor lines. The authors report that evolutionary young LINE elements (and some other types of evolutionary young retrotransposon) are particularly affected by hyperexpression/de-repression after loss of DNA methylation. This affects also the expression of disease-relevant protein-coding genes that are in proximity of LINE1 elements, including numerous neurodevelopmental risk genes.

The novelty of this paper is two-fold (i) the phenotype is milder than in DNMT1 mouse 'knock-out cells and (ii) the massive loss of DNA methylation after DNMT1 loss of function mutagenesis primarily derepresses the evolutionary younger type of repeat elements.

The paper appears to be improved since initial submission to Nature journal series, for example the additional data shown in figure 1d-f, documenting massive loss of methylation by WGBS method provide convincing evidence that the mutant neural progenitor cells indeed are affected by massive DNA methylation loss.

We thank the reviewer for the overall positive comments on our study. In the new version of the manuscript we have added new experimental data and also made several changes to the discussion of the results. We now hope that the reviewer agrees with us that the manuscript is ready for publication.

Major comments:

1. The most exciting question in the field (does derepression of young LINE1 elements increase the number of de novo insertion sites with somatic mutagenesis in these neural progenitors) is not addressed in this paper.

We agree with the reviewer that we not address the question of retrotransposition in somatic cells in this manuscript. During the last decade there has been numerous studies that use different methodologies to prove the existence of somatic transposition events in NPCs and the brain. Despite this huge effort by several world-leading labs, the data remain controversial. The reason for this is that the methodology to prove somatic retrotransposition events is extremely challenging (and probably still premature). Given this, we feel that this type of analysis lies well outside the scope of the current study. Our current study should rather be seen as an investigation into the epigenetic control of TE-expression – which obviously could result in retrotransposition but may also have other consequences.

2. Because evolutionary older repeat elements have a much higher burden of

degenerative mutations, rearrangement etc , isn't it expected that an epigenetic alteration such as loss of DNA methylation primarily results in derepression of the younger type of repeat elements. This is not properly discussed , I feel.

We agree with the reviewer that the evolutionary age of a TE will influence the probability to be expressed due to the mutational burden. While this certainly influence our observations, it is not the only explanation. For example, SVAs and many HERVs, which are young elements with the potential to be expressed (as demonstrated by many other reports) remain silent despite the lack of DNA-methylation. We have clarified this issue in the discussion, p. 14.

3. Some important comments raised in previous review, including the recommendation to confirm the acclaimed 'massive increase in levels of L1ORF protein' by western blot , are still unaddressed.

To address this issue, we have performed western blot analysis of L1-ORF1 protein. This analysis demonstrates expression of L1-ORF1p in *DNMT1*-KO NPCs but not ctrl NPCs. This analysis has been inserted as Supplementary Fig. 1p and mentioned on p. 9, as well as in Supplementary Methods.

Reviewer #5 (Remarks to the Author):

Jönsson et al have generated DNMT1 KO hNPC cells, which surprisingly remain viable, and found pronounced upregulation of young L1 elements, some of which generate chimeric transcripts with neuronal genes.

These are key findings when it comes to understanding the role of DNA methylation and other repressive mechanisms in TE (and gene) regulation in differentiated tissues, about which we know so little. The data backing the dependency of young L1 repression on DNMT1 are extensive and robust, which includes a careful navigation of the caveats associated with mapping of short read data to TEs. The observations on chimeric transcripts are also of interest, although the link between L1 demethylation and gene activation in neuronal differentiation and/or neurodevelopmental disorders is unclear.

We thank the reviewer for the overall positive comments on our study and the useful comments. We have now generated a new version of the manuscript that takes these suggestions into consideration.

I have the following suggestions for the authors' consideration:

1) Is there evidence from RNA-seq data that the L1 fusion transcripts identified here are expressed upon hNPCs differentiation and/or in neurodevelopmental disorders? This would provide a stronger link with a physiological context.

We agree with the reviewer that this is a very important point. To start addressing

this issue we have generated new high-quality RNA-seq data from human brain development (using tissue from aborted fetuses). This analysis demonstrates that some of the L1-fusion transcripts can be detected during human brain development. This new data is inserted as Supplementary Fig. 3e and mentioned on p.13 in the results section.

When it comes to neurodevelopmental disorders we agree that this is extremely interesting and we currently working on this topic in the lab. However, to properly address this question extensive in vitro modeling (using e.g. patient derived iPSCs) and analysis of post-mortem material will be required. We feel that this lies well outside the scope of the current study.

2) Although L1s are the most prominently affected TEs in the KO, some LTRs are also greatly upregulated. Notably, many copies of LTR12C elements are derepressed, and these have also been previously linked with the generation of chimeric transcripts (e.g., PMID: 28604729). It would be interesting to know to what extent this LTR family is also generating chimeric transcripts in hNPCs and, if so, how they compare with the ones from L1s with regards to gene ontology and expression patterns.

We thank the reviewer for pointing this out. We agree that the observations regarding LTR12C are interesting. We have now added a new Supplementary figure that contains more detailed expression on LTR-elements upon DNMT1-KO as well as analysis of fusion transcript. This new data is inserted as Supplementary Fig. 1o and 3c and mentioned on p.8 and 12 in the results section.

3) The authors acknowledge in the Methods section that polymorphic L1s can affect the outcome of sequencing data mapping. This should probably be mentioned in the main text, as it is an important point.

In line with this suggestion we have moved these statements to the results section, page 8.

The following are non-essential points that mainly stem from my curiosity about the data:

4) I cannot help wonder what happens to H3K9me3/TRIM28 at different L1 families in the DNMT1 KO cells. It would be great if the authors had some ChIP-qPCR data on this.

We agree with the reviewer. However, since we generate the DNMT1-KO cells in bulk we need to perform the ChIP on quite few cells. Unfortunately, we have not been able to do a successful H3K9me3 ChIP with this cell number.

5) I take it from the GO analysis in Figure S1h (and from cell survival) that L1 activation does not lead to an interferon response, as seen elsewhere. This may be worth mentioning.

This is an important point. It is true that the GO analysis does not indicate an interferon response. We have tried to do this analysis with many different cut-offs and it never brings out an interferon response. However, we are not completely convinced that there is a complete absence of this phenomenon. For example, we have looked in detail at a panel of 24 viral defense genes described by Peter Jones and co-workers (PMID:30185548) and we find that 6 of these genes are upregulated upon DNMT-KO. Thus, there might be a weak response, or we may be looking at the wrong time point. Given that these observations still are inconclusive we have decided to leave this analysis out of the current manuscript. We are currently investigating this in detail in the NPCs (as well as in brain cancer cells) but this analysis is still premature and clearly outside the scope of this study.

6) Do the authors have any data on whether the piRNA pathway becomes active upon DNMT1 KO?

We find that several genes in the piRNA pathway are activated upon DNMT1-KO. However, we have not performed an analysis to investigate if piRNAs are being produced.

7) Do DNMT1 KO hNPCs die upon differentiation?

We have tested to differentiate the DNMT1-KO cells and they do not form neurons using the standard differentiation protocol for these cells. If they die or just don't differentiate is difficult to assess. We have also no mechanistic explanation to this observation. We therefore feel that this observation lies outside the scope of the current study and that a substantial detailed analysis would be required to pinpoint what is going on. We have therefore left this observation outside the current manuscript.

REVIEWERS' COMMENTS:

Reviewer #4 (Remarks to the Author):

The Reviewers have addressed all my previously raised points. The newly added western blot data on L1 ORF1P in Supplementary Figure 1 are convincing.

Although the issue of increased insertional mutagenesis /increased L1 retrotransposition is still not addressed, the Authors have a point in arguing that this maybe beyond of the scope of this manuscript.

Overall the paper is now further improved and I have no additional concerns.

Reviewer #5 (Remarks to the Author):

New data has been included that addresses all of my main questions to the authors.

Miguel

Reviewers' comments:

Reviewer #4 (Remarks to the Author):

The Reviewers have addressed all my previously raised points. The newly added western blot data on L1 ORF1P in Supplementary Figure 1 are convincing. Although the issue of increased insertional mutagenesis /increased L1 retrotransposition is still not addressed, the Authors have a point in arguing that this maybe beyond of the scope of this manuscript. Overall the paper is now further improved and I have no additional concerns.

We thank the reviewer for the comments.

Reviewer #5 (Remarks to the Author):

New data has been included that addresses all of my main questions to the authors.

We thank the reviewer for the comments.